# Improving Learning Conditions for Computer Science Students by Using the Flipped Classroom

## Abstract

The flipped classroom is a pedagogical strategy that provides the instructors with a method of minimizing the amount of traditional direct instructions during the teaching process while emphasizing the one-to-one interaction. The students watch some educational resources before attending the session in which they build the required knowledge and prepare questions to be discussed during the session. Such resources are accessed via technological tools, offering flexibility and accessibility. The flipped classroom offers self-paced learning process supporting mastery learning and self-efficacy.

In order to improve the learning conditions of Computer Science students at the Lebanese University - Faculty of Sciences - Fanar, we tested the flipped classroom in two master's courses. We designed these courses based on the ADDIE pedagogical model, we used different tools to set up our teaching-learning activities and we evaluated the work of our students by the rubrics-based evaluation method. We prefer to apply the flipped classroom in other courses, starting next academic year, to assess the student's satisfaction.

**Keywords:** flipped classroom, e-learning, teaching method, pedagogical scenario, pedagogical model.

## 1 Introduction

The quality of education is the main target of the educational systems. Educational strategies and methods are improved and enhanced continuously and regularly in order to facilitate the learning process. E-learning is an educational system that uses technological sources to provide an effective and efficient learning environment, regardless of time and location. The E-learning helps the students to obtain productivity by which they do their tasks and assignments efficiently Rahmelina et al. (2019). To maintain quality learning, significant improvements for technological resources as well as users are required. This has opened the door for teachers to alter the way they deliver the lectures. This changing and transformation in teaching method require the integration of a new pedagogical approach which is the flipped classroom. Flipped classroom is a pedagogical model and learning strategy altering and reorganizing the traditional learning environments. It integrates the technological resources as a support to help students navigate the online material. The material is first delivered as e-learning providing training and support videos, as well as using group discussion and assignments. In this case, the teachers play the role of advisors to direct the students to the right path of learning. The integration of flipped classroom in e-learning environments has offered irreplaceable advantages in the modern teaching. For example, it has encouraged the students to watch videos, access the material given and perform their assignments while forming the concept of learning done at home. The term "flipped" comes from the concept of turning homework into classwork. The revised Bloom's Taxonomy of the cognitive learning domain Krathwohl (2002) is an excellent presentation that supports this concept Nechodomu et al. (2016), it is illustrated in Figure 1.

In traditional learning, the two lower levels of learning, remembering and understanding, take place in class, the students must work on activities on higher levels of learning outside the classroom. However, in the flipped classroom concept, learning is flipped, so students can finish the two lower levels of cognitive work before attending the course and can participate in both levels of cognitive

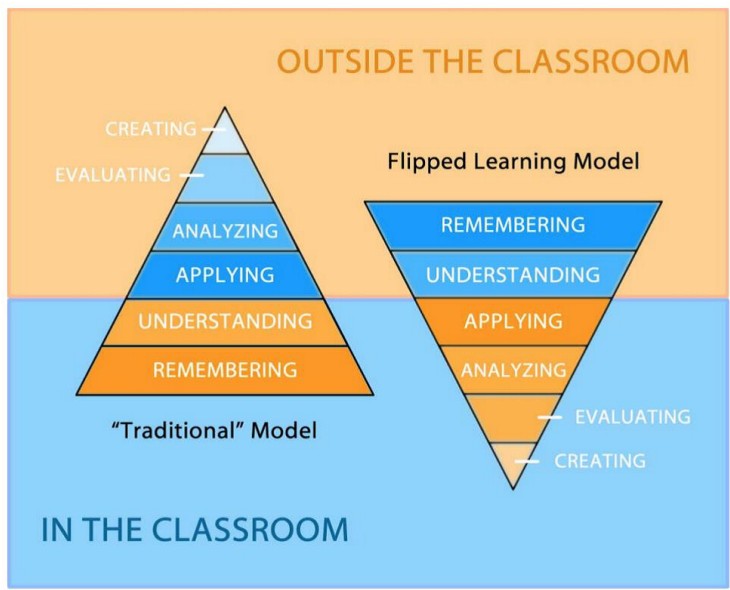

Figure 1: Revised Bloom's Taxonomy and flipped classroom approach.

learning, applying and analyzing, during the class, with their peers in the presence of the teacher. Then they can start the two highest levels after class (Figure 2) Anderson & Krathwohl (2001).

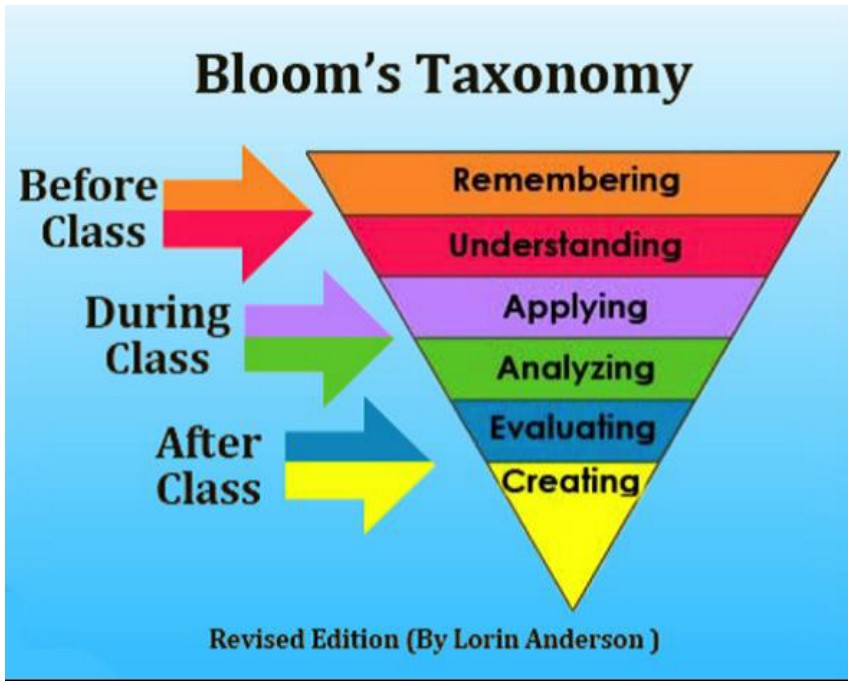

Figure 2: Activity levels before, during and after class.

The flipped learning model using e-learning media facilitate the student's learning process because of its effectiveness Rahmelina et al. (2019). Upon the efficiency and effectiveness of flipped classroom, teachers are implementing video tutorials as their teaching method Ash (2012).

The concept of the flipped classroom is to build a learning environment under the light of technological advances and focusing on the student-centered approach Yilmaz (2017).

The traditional classroom learning environment that focuses on the transmission of knowledge from the teacher to the student decreases the student's interest and ability to comprehend after an hour. This results in lowering the probability of student's engagement with the classroom content. Therefore, a new pedagogical method is required to overcome this issue. The flipped classroom provides the students some autonomous work through the integration of technological resources before the class and by practical activities during the classroom in order to build and strengthen the student's knowledge. The flipped classroom effects the active learning because it is a critical methodology to enhance the active construction of knowledge, fostering a fruitful environment of interaction between the students, and between them and the teachers. The active learning environment has facilitated the student's learning process by acquiring knowledge at home, and then implementing this knowledge individually in the class. Since the students engage with each other by sharing their experiences and knowledge, the flipped classroom enhances the sense of belonging to the community. This results in an opportunity to develop their communication skills and from different perspectivesSantos & Serpa (2020). The video lecture is seen as a key factor for the flipped classroom which typical lecture and homework elements are reversed. These short videos serve as a tutorial in which the students move at their own pace, review options and skip the sessions they are familiar with. These videos are designed specifically to suit each student by means of providing captions, familiar languages Elazab & Alazab (2015).

To provide our students with pedagogical continuity following the health crisis resulting from the Covid-19 pandemic and the unstable economic situation, we asked the following question: how can we improve the learning conditions for the student?

To answer this question, the aim of our work is to design the scenario of a teaching sequence focuses on the student according to the flipped classroom approach, to identify the relevant technological tools in order to create the resources, manage the learning activities, and finally to evaluate the acquired knowledge of the students following a teaching sequence.

We applied this approach at Lebanese University - Faculty of Sciences - Fanar, on two courses (Mobile Application and Network) for first-year master's students of Computer Science.

The remainder of this paper is organized as follows: the flipped classroom effectiveness is described in Section 2, the teachers' perspective and the students' perception regarding flipped classroom are tackled in Sections 3 and 4, an overview about the challenges of the flipped classroom is presented in Section 5, the methodology of our work is described in Section 6, and finally Section 7 concludes this paper and presents our future perspectives.

## 2 FLIPPED CLASSROOM EFFECTIVENESS

Students have different learning or cognitive styles in acquiring and building their knowledge. Studies show that the flipped classroom is a positive learning strategy as it increases the student's performance. Watching and listening to recorded lectures is useful and promotes the student's activity. The flipped classroom enhances the sense of responsibility among students of while watching the lectures to build their knowledge. In addition, students feel that they have more control as they have to prepare themselves for the upcoming session in a more active way than the traditional classroom's. This has resulted in a sense of empowerment and control among the students. Furthermore, the flipped classroom has offered mobility, flexibility and accessibility to watch the lectures Avdic & Åkerblom (2015). Flipped classroom offers repetition through the ongoing access to educational material, collaboration between students and teachers, and flexible preparation. The implementation of flipped classroom has altered the student's performance positively. In addition, it has various advantages in improving interactions, developing skills, and engaging the learner in the learning processJones-Bonofiglio et al. (2018). The students' understanding of basic concepts is more effective during active engagement in learning rather than the passive environment. In addition, the students believe that the flipped classroom helped them in taking notes easily at their own pace, watching the missed sessions, and having a reference. Students have more time and pace to interact with the teachers during the session rather than being passively listening to the lecture. Moreover, the flipped classroom promotes mastery learning, which focuses on ensuring that the level of mastery is

met before moving to another set of learning outcomes. Students acquire the basic knowledge before attending the session, and after that the teacher assesses the student's knowledge and performs corrective tutoring in order to ensure learning mastery. Consequently, mastery learning has shown positive impact on the student's abilitiesJohnson (2013). The flipped classroom concentrates on transforming the student from a passive role into an active role, learning and teaching regardless of place and time, and designing content focusing on the purpose of the lectureColomo-Magaña et al. (2020).

## 3  TEACHERS' PERSPECTIVE REGARDING FLIPPED CLASSROOM

Teachers have different perspectives regarding the flipped classroom implementation and effectiveness. Although video tutorial materials are used in the learning process, yet if the lecture is still the primary mode of learning, the shifting into flipped classroom is not accomplished. Other instructors argue that the styles of learning of students differ; some might find the videos helpful while others won't even watch. Some used the flipped classroom as a supplementary style to provide more instructions to students after basically explaining the base concepts to them. Another strategy used was to post educational videos and then ask students to take notes and come up with questions, thus enhancing the learning process through sharing and discussing information. Instructors emphasized that the flipped classroom provided a wide framework of instructional methods and styles to reach students and help in their learning process Ash (2012). The flipped classroom environment provides an optimal and efficient use of the instructors as well as students' time. The flipped classroom provides an effective classroom time to display the intended content, open discussions regarding the topics and offers interaction between the students. As students have access to the content online, less time is spent in answering basic and repetitive questions. In addition, the recorded lectures are being used years over years for their easy updating process of their content and fast adaptation to respond to new learning styles and demands. Such lectures enhance the student's understanding of the material, facilitate the students' preparation for exams and tests, and increase the test performance of the students Elazab & Alazab (2015). In addition the teachers' responsibility is more focused on understanding the student's knowledge structure in order to develop a suitable content Santos & Serpa (2020).

## 4  STUDENTS' PERCEPTION REGARDING FLIPPED CLASSROOM

One of the factors that effects the student's motivation and satisfaction in the flipped classroom model is the readiness of the students for the e-learning environment. The first step in the flipped classroom courses is the successful completion of the online requirements which is contributed to the student's satisfaction as well as to his motivation to pursue the flipped classroom activities. Nevertheless, the success of the flipped classroom courses is associated with intra-personal skills such as self-efficacy, self-regulation, time management, communication skills as well as goal directed behaviors. These skills alter the student's perception of the e-learning experience and thus impacts the satisfaction. Another factor that affects the readiness of the student is the desire having physical, mental and effective components ensuring the readiness conditions for e-learning. Studies have shown that forcing students for an e-learning experience results in a negative experience and perception for future experiences and similar activities. The level of satisfaction is also affected by the habits the students are used to from previous traditional classes. In addition, students in flipped classroom model have to watch instructional videos before going to class. This represents additional work for students that require great motivation. Therefore, motivation is the most critical element of the structural design of the instructional environment. The learning experience is considered unsuccessful unless the students are motivated to complete the course. Students have habits from the previous traditional experiences, thus switching into online learning environment requires motivation to engage into. Taking into account that students nowadays are technologically natives, however, online learning environment is a complex one. Therefore, improving the self-efficacy impacts the students' satisfaction during the online experience. Studies have shown that students' interaction, with other students, content or with the teacher, is a major factor that determines student's satisfaction Yilmaz (2017). The flipped classroom has shed a greater responsibility on the students with greater work and motivation, thus making it impossible to play the role of a passive learner in such environment. The students play a critical and active role, giving more satisfaction Santos & Serpa (2020). In a

study focusing on the flipped classroom outcomes, students have reported gaining new knowledge, engaging more in the learning setting as their confidence has increased, and meeting the course objectives Jones-Bonofiglio et al. (2018). The study was carried out in order to evaluate the perception of students regarding the implementation of flipped classroom shows that the students have positive perception. The students agreed that flipped classroom promotes learning Colomo-Magaña et al. (2020).

## 5 FLIPPED CLASSROOM CHALLENGES

Although the flipped classroom has offered flexibility regarding the lectures' accessibility, some students have reported that self-discipline is necessary in the learning process. It is easier for students to follow a specific schedule rather than allocating time according to their preferences. Moreover, students feel the need to ask questions while watching the lectures, which is impossible during the recorded lectures. Since the flipped classroom is associated with more responsibility among the students, it created a source of discomfort. The unfamiliarity with the flipped classroom added a burden on the student's learning process Avdic & Åkerblom (2015). Studies have shown that the psychological relationship between the teacher and learner affects the students' performance and satisfaction as it plays a role in establishing a sense of safety. Therefore, students have to create a psychological relationship with the teacher before initiating the learning process Jones-Bonofiglio et al. (2018). Furthermore, students feel more stressed since they play the leading role in the learning process.

## 6 METHODOLOGY

To apply the flipped classroom, we started to design our course using the pedagogical engineering model, ADDIE Molenda (2015), which is based on the following five consecutive steps: Analysis, Design, Development, Implementation and Evaluation.

1. Analysis: In this step, we, as teachers, have analyzed pedagogical needs. Since the student is the center of the learning system, we have studied their technological, educational, and material needs. An analysis of the resources was made, the working environment and the necessary means.

2. Design: in this part we have determined the teaching method and its pedagogical scenario. We divided this educational scenario into two main parts. The first part describes the general presentation model of the course. This model is called macro design and includes the following items: Course title, General description, Target audience, General objectives, Targeted skills, Prerequisites, Duration, Teaching method (face-to-face, interactive sessions, practical work, and tutorials), and Language (French or English). The second part is dedicated to the planning of each teaching-learning activity such as exercises, practical work, tutorials and project. This part is called micro design, it is presented in Table 1.

   The term "Synchronous" means that the activity is done by the student in the presence of the teacher in class or online. The term "Asynchronous" means that the activity is done without the presence of the teacher, the student works alone or in a group. The objective is an action verb, if it is correctly defined, it is half achieved. The content is only a means. For this, in both parts, the objectives are defined from the revised Bloom's taxonomy Krathwohl (2002) which targets cognitive type learning (this is the type of our learning). Bloom's taxonomy makes it possible to classify educational objectives into six levels and types of skills (Remember, Understand, Apply, Analyze, Evaluate, and Create) and for each level, a series of action verbs is proposed.

3. Development: In this part, we have developed the content of our course and we determined the teaching aids to be used. To prepare a pedagogical scenario for our course (including exercises and problems), we used the Learning Design tool. The content of our course (enriched by pedagogical multimedia) was based on the principles of Mayer Mayer (2008) and on his three important foundations for multimedia learning:

   - Dual channels: humans possess separate channels for processing visual, sound, and verbal material.

Table 1: Planning teaching-learning activities

| | MODULE: mobile cross platform frameworks | | |
| --- | --- | --- | --- |
| | Before class (asynchronous) | During class (synchronous) | After class (asynchronous) |
| Steps/Timing | -Consult videos and websites / 12 mins -Collaborate together (group of 2) to summarize their research / 60 mins | -Present the research summaries / 30 mins -Exercises (practical work + tutorials) / 60 mins | -Develop a small mobile application project / 80 mins |
| Objectives | -Know how to find the right resources | -Practice the use of different functions | -Design mobile applications |
| What the student does (Alone/in a group) | -Consult (alone) -Search (alone) -Collaborate (in groups) -Summarize (in groups) | -Participate in tutorials (in groups) -Solve the practical exercises (alone) | -Work together (group of 2) Develop the project |
| What the teacher does | -Put resources -Provide websites -Be available to students | -Answer the questions -Guiding the students -Correct answers | -Support students |
| Tools | -Documents -Internet | -Software | -Internet -Software |
| Resources | -Videos -Websites | -Presentation -Documents | -Documents |

- Limited capacity: each channel can process only a small amount of material at any one time.
- Active processing: deep learning depends on the learner's cognitive processing during learning.

Other tools have been made available to us, such as:

- Padlet: collaborative wall that allows students to introduce themselves in a few words and share their ideas in a few lines.
- Quiz-Maker: allows the teacher to create a set of short and effective quizzes to test the knowledge of his students, to check if they have studied the course and understood the educational activity. The teacher can also add a score to each quiz.
- Answergarden: allows students to answer more elaborate and detailed exercises proposed by the teacher.

4. Implementation: we have implemented and applied the flipped classroom in two courses, Network and Mobile Application, for Master Computer - Science students.

5. Evaluation: students must participate in graded practical work. These are presented in the form of problems to be solved in class or at home and in the form of a project to be presented when the course is finished. To evaluate the work of the students, we used a rubrics-based evaluation which is only of interest in the context of the use of open evaluation methods such as oral presentation, group project, etc. Dominguez et al. (2019). In this grid, our evaluation criteria are as follows:

- The presentation of the work must meet the quality criteria.
- The student's written and oral expression must not contain any syntax or spelling errors.

- The vocabulary must be varied.
- Answers and information must be relevant to understand the subject presented.
- Finally, the student's ability to carry out research and provide a summary on the subject is assessed.

Each criterion is evaluated by one of the following terms: Excellent, Very Good, Satisfactory, or Fair.

## 7 CONCLUSION

The flipped classroom pedagogical method encompasses an innovative and transformative teaching model, shifting the classroom from teacher-centered into student-centered environment. The student is considered a real player in flipped Classroom. However, the success of the flipped classroom environment is impacted by several factors including the adaptation and preparation of the teachers as well as the student's satisfaction. If this teaching-learning system is not well designed and developed, it will result in learning difficulties among the students and teaching difficulties among the teachers in transferring knowledge. However, if the flipped classroom method is well implemented, it enhances the learning potential of the students. To model such teaching and integrate it into the educational program, we followed the ADDIE pedagogical engineering model. We applied two models, macro and micro design, for the scenario of the educational activity. We used several tools to enrich our teaching. Our objective, which is to improve the student's learning conditions by offering him an education based on the flipped classroom, complementary to his face-to-face teaching and accessible without time or space constraints, has been achieved, but this is just a first step. To continue our work, certain perspectives are already envisaged. Indeed, we tried and tested the flipped classroom only in two courses. For a better continuity in its use, it must be applied in many courses of the four years of studying Computer Science at the Lebanese University - Faculty of Sciences - Fanar. When all the courses are scripted, the flipped classroom becomes a veritable library of digital resources at the student's service. This will also strengthen the regular follow up of the students throughout their learning. In this perspective, many other teachers must be trained to use and apply this method.

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
