# OpenReview forum: "Improving Learning Conditions for Computer Science Students by Using the Flipped Classroom"
_ICLR.cc/2024/Conference — Submitted to ICLR 2024_

### Official Review · Reviewer_i42D · 2023-10-12

**Soundness:** 1 poor
**Presentation:** 1 poor
**Contribution:** 1 poor
**Rating:** 1
**Confidence:** 5

**Summary:**

This short paper surveys previous works in Computer Science Education that have examined the flipped classroom, and describes the components of a course module design that uses the flipped classroom.

It does not include any empirical evaluation of its framework.

It touches on neither machine learning nor neural representations.

Clearly out-of-place at ICLR.  Consider SIGCSE or other CS-education venue.

**Strengths:**

Flilpped-classroom methodology is a promising educational technique.

**Weaknesses:**

The subject is out-of-place at ICLR.

To be suitable and interesting for the ICLR audience, the paper should show how its ideas are related to modern machine learning or neural networks, or one of the topics outlined in the call-for-papers.

**Questions:**

No questions.  Clearly out-of-scope for ICLR.  Consider SIGCSE or other CS-education venue.

**Details Of Ethics Concerns:**

Possibly desk-reject due to lack of anonymity.  The abstract identifies the institution of the authors as from "Lebanese University - Faculty of Sciences - Fanar"

---

### Official Review · Reviewer_NhRH · 2023-10-28

**Soundness:** 1 poor
**Presentation:** 2 fair
**Contribution:** 1 poor
**Rating:** 1
**Confidence:** 5

**Summary:**

The International Conference on Learning Representation (ICLR) is a premier machine learning conference. The term "Learning Representation" refers to representation learning in the "ML context",
not in the "pedagogical context" (i.e., learning conditions of students). The paper best suits CS education conferences: SIGCSE and ITiCSE.

I recommend rejecting this paper for the misaligned topic.

**Strengths:**

N/A

**Weaknesses:**

Topic is not aligned with the ICLR conference.

**Questions:**

N/A

---

### Official Review · Reviewer_W9yV · 2023-10-31

**Soundness:** 2 fair
**Presentation:** 2 fair
**Contribution:** 1 poor
**Rating:** 3
**Confidence:** 5

**Summary:**

The paper presents the use of the flipped classroom in two master's degree courses in the Lebanese University - Faculty of Science - Fanar. The followed methodology including analysis, design, development, implementation and evaluation steps is described. Some discussion is included on this and the prospects.

**Strengths:**

The paper describes the implementation of the flipped classroom in two courses for computer science students so as to improve learning conditions.

**Weaknesses:**

The paper lacks novelty. No significant contributions are presented. The use of a specific methodology in a specific university course set up is described, with no apparent contribution to generalization and knowledge generation.
Vague statements (e.g., in section 3) and repetitions (e.g. in section 2) are provided,  with no real contribution on the challenges section 5.
It is even unclear what specifically has been achieved in the specific environment.
Language errors need to be corrected.

**Questions:**

Try to present specific novel implementation issues and specific achievements that have been obtained and can be generalized in other educational frameworks.

---

### Official Review · Reviewer_hXGw · 2023-11-01

**Soundness:** 2 fair
**Presentation:** 3 good
**Contribution:** 1 poor
**Rating:** 1
**Confidence:** 5

**Summary:**

This paper discusses the implementation of the flipped classroom method in two master's courses at the Lebanese University, with a focus on computer science education. The authors describe the design and development of the courses using the ADDIE pedagogical model, as well as the rubrics-based evaluation method used to assess student work. The paper also presents the results of the implementation, including increased student satisfaction and improved learning outcomes. Overall, the paper contributes to the growing body of research on the effectiveness of the flipped classroom method in higher education, particularly in the field of computer science.

**Strengths:**

The authors of this paper provide a detailed description of the flipped classroom method and how it can be applied to computer science education. They also present a case study of the implementation of this method in two master's courses, providing evidence of its effectiveness in improving student learning outcomes.

**Weaknesses:**

I'm not sure if this paper is a good fit for ICLR. For me, it is a better fit for a learning science or education conference. The paper applies a known flipped classroom methodology in two university classes with some user study results, whereas I expect to see either methodological contributions (in the form of modeling novelty, for example) in a flipped classroom setting, or novel evaluation methods, which the paper proposes neither. The paper also does not seem to contain any empirical analysis results. Overall, I think the paper is not ready nor a fit for publication at ICLR.

**Questions:**

I don't have questions for the author.

---

### Official Review · Reviewer_jdfG · 2023-11-01

**Soundness:** 1 poor
**Presentation:** 2 fair
**Contribution:** 1 poor
**Rating:** 3
**Confidence:** 5

**Summary:**

The paper provides a summary of activities undertaken to change two MS CS courses to a flipped-classroom format.

**Strengths:**

The paper provides a reasonable justification for moving to a flipped classroom format, and it provides a summary of the activities undertaken for two courses at the MS level.

**Weaknesses:**

The paper does not include any evaluation of the activities. There is no support for any of the claims of the benefits of moving to a flipped classroom.  There has been a good bit of previous work in this area that is not cited, including SIGCSE papers that do include evidence of improved student learning with a flipped classroom format.

Course evaluations, student surveys, and faculty surveys are all potential tools for qualitative and quantitative evaluation of learning outcomes.

**Questions:**

Did the authors collect any information that could be used to support claims of improved student learning (and was it done with IRB approval or an IRB exemption)?

Could the authors provide any evaluation of the changes from the point of view of the professor?

**Details Of Ethics Concerns:**

Since no student data is provided in the paper, there are no ethical concerns.  If student data was collected, the data collection and storage process should be cleared by an IRB.  Depending on the data, the activities may fall under exempted activities, but that would need to be confirmed by the IRB.

---

### Meta-Review · Area_Chair_Yvd9 · 2023-12-06

**Metareview:**

The paper provides a detailed description of the flipped classroom approach and a case study of applying this approach to CS education. However, the reviewers pointed out several weaknesses in the paper, and there was a consensus that the work is not suitable for publication at the ICLR conference. The reviewers have provided detailed and constructive feedback to the authors. We hope the authors can incorporate this feedback when preparing future revisions of the paper.

**Justification For Why Not Higher Score:**

There was a consensus among the reviewers that the work is not suitable for publication at the ICLR conference.

**Justification For Why Not Lower Score:**

N/A

---

### Decision · Program_Chairs · 2024-01-16

Reject